# Multi-Feature Matching GM-PHD Filter for Radar Multi-Target Tracking

**DOI:** 10.3390/s22145339

**Published:** 2022-07-17

**Authors:** Jin Tao, Defu Jiang, Jialin Yang, Chao Zhang, Song Wang, Yan Han

**Affiliations:** Laboratory of Array and Information Processing, Hohai University, Nanjing 210098, China; taojin@hhu.edu.cn (J.T.); 210407080001@hhu.edu.cn (J.Y.); 200207080004@hhu.edu.cn (C.Z.); 160407080001@hhu.edu.cn (S.W.); hyan0525@hhu.edu.cn (Y.H.)

**Keywords:** radar, multi-target tracking, RFS, GM-PHD, multi-feature matching

## Abstract

Multi-target tracking (MTT) is one of the most important functions of radar systems. Traditional multi-target tracking methods based on data association convert multi-target tracking problems into single-target tracking problems. When the number of targets is large, the amount of computation increases exponentially. The Gaussian mixture probability hypothesis density (GM-PHD) filtering based on a random finite set (RFS) provides an effective method to solve multi-target tracking problems without the requirement of explicit data association. However, it is difficult to track targets accurately in real-time with dense clutter and low detection probability. To solve this problem, this paper proposes a multi-feature matching GM-PHD (MFGM-PHD) filter for radar multi-target tracking. Using Doppler and amplitude information contained in radar echo to modify the weights of Gaussian components, the weight of the clutter can be greatly reduced and the target can be distinguished from clutter. Simulations show that the proposed MFGM-PHD filter can improve the accuracy of multi-target tracking as well as the real-time performance with high clutter density and low detection probability.

## 1. Introduction

### 1.1. Background and Motivation

With the capability of all-weather monitoring day and night, radar has been widely used in civil and military applications [1,2,3]. MTT is one of the most important functions of radar systems. With the development of sensor technology, radar has been used in complicated scenarios [4,5]. In radar MTT applications, the key challenges encountered involve data association uncertainty, detection uncertainty, clutter, false alarms, and real-time performance. Therefore, real-time and accurate multi-target tracking has become a critical issue in the field of radar MTT.

Traditional MTT technology based on data association, such as multiple hypothesis tracking (MHT) [6], joint probabilistic data association (JPDA) [7], and probabilistic multiple hypothesis tracking (PMHT) [8], convert multi-target tracking problems into single-target tracking problems by matching measurements with targets. When the number of targets and false alarms is large, combination for matching would increase the computational complexity exponentially. Mahler proposed a random finite set theory to solve the traditional data association problem of multiple-target tracking [9,10]. In the RFS formulation, the collection of individual targets is treated as a set-valued state, and the collection of individual observations is treated as a set-valued observation. The probability hypothesis density (PHD) filter was proposed as an approximation to the RFS solution for MTT. Two different implementations of the PHD filter, the sequential Monte Carlo PHD (SMC-PHD) [11,12,13] and the Gaussian mixture PHD [14,15,16], are among the different PHD filter methods. Compared to the SMC-PHD filter, the GM-PHD filter has the advantages of a low computational cost and simple state extraction. However, when the clutter density increases and the detection probability declines, the performance of the basic GM-PHD filter degrades significantly. Therefore, the focus of this paper is to improve the tracking accuracy and real-time performance of GM-PHD filter in a dense clutter and low detection probability environment.

### 1.2. Brief Survey of Related Work

In radar applications, tracking accuracy and real-time processing are important. In dense clutter environments, the GM-PHD filter has difficult with accurate multi-target tracking. In order to improve tracking accuracy performance, [17] proposed the generalized label multi-Bernoulli (GLMB) filter and [18] proposed the label multi-target Bayesian processing method, which significantly improved the accuracy of multi-target state extraction. Subsequently, [19] proposed an efficient implementation of the GLMB filter by integrating the prediction and update into a single step and improve both the cardinality estimation and the state estimation by truncating the GLMB filtering density based on Gibbs sampling. However, these algorithms improve tracking accuracy at the cost of higher computational complexity. Therefore, [20] proposed a labeled GM-PHD filter which eliminates the negative effects of clutter by the identity label of the individual target and improves real-time tracking performance significantly. However, in high clutter rate and low detection probability scenarios, tracking accuracy of this method declines greatly. Feature-aided tracking methods which attempt to extract features from data for multi-target tracking have been used in video [21], satellite video [22], and high-resolution remote sensing imagery [23]. A feature-aided extended target probability hypothesis density filter for high resolution radar using the number and spatial extension of measurements as features to deal with extended target tracking was proposed by [24]. A feature-aided SMC-PHD filter for nonlinear multi-target tracking using the signal-to-noise ratio (SNR) and down-range extent (DRE) as features to increase the estimation performance of the SMC-PHD was proposed by [25]. All these methods effectively improve the target tracking performance using feature information embedded in measurements.

Radar echoes contain a lot of feature information, such as range, angle, amplitude, and Doppler information. Due to the different features of targets and clutter, it is possible to use these features to improve tracking performance. In this paper, we propose a multi-feature matching GM-PHD filter, using Doppler and amplitude information to modify the weights of the Gaussian components. The clutter is reduced through prediction and update process. After pruning and merging, clutter can be further eliminated and the merging of different targets can be prevented through the utilization Doppler and amplitude information. Simulations show that the proposed MFGM-PHD filter has good tracking accuracy performance and real-time performance in dense clutter and low detection probability environments.

### 1.3. Main Contributions

The major contributions of this paper are as follows:

First, a multi-feature matching GM-PHD filter is proposed to effectively track targets with high clutter density and low detection probability for radar.

Second, the weights of Gaussian components are modified using Doppler and amplitude information contained in radar echo so that the weight of clutter can be greatly reduced. Thus, the tracking accuracy performance and real-time performance are improved. The performance of the proposed approach is verified by simulations.

Third, during the pruning and merging procedure, clutter can be further eliminated and the merging of different targets can be prevented using Doppler and amplitude information.

The rest of this paper is organized as follows: Section 2 contains a brief background on radar MTT and PHD filters based on RFS. Section 3 presents the multi-feature matching GM-PHD Filter. The simulation results are given in Section 4, followed by the conclusions of this study in Section 5.

## 2. Background

### 2.1. Radar Multi-Target Tracking

Radar systems generally consist of a signal transmitter, a signal receiver, signal processing components, data processing components, and other important parts. As shown in Figure 1, the signal transmitter and receiver are responsible for sending and receiving electromagnetic waves. Signal processing extracts target measurement data according to the characteristics of transmitted and received signals [26]. Data processing estimates the target state based on measurement data, and target tracking is the main method for target state estimation [27]. As the main function module of data processing, the target tracking module is responsible for estimating specific target information such as position, velocity, and acceleration according to the output data of signal processing, so as to realize the target detection function of radar.

Traditional radar multi-target tracking methods adopt data association. These methods transform the multi-target problem into a parallel single-target problem. However, when there are many targets and a large amount of clutter, the performance of these traditional methods is significantly reduced and the computational intractability far more severe. Therefore, the PHD filter operates on the single-target state space and avoids the combinatorial problem that arises from data association [16]. These salient features render the PHD filter extremely attractive.

### 2.2. PHD Filters Based on RFS

In RFS-based filters, the multi-target states and multi-target measurements are modeled as finite sets. Suppose there are M(k) targets at time k. The multi-target state is represented as
(1)$$X_{k}=\{x_{k}^{1},x_{k}^{2},\ldots ,x_{k}^{M(k)}\}\in F(X)$$
where $x_{k}^{1},x_{k}^{2},\ldots ,x_{k}^{M(k)}$ represent the state vectors of M(k) targets respectively and F(X) represents all subsets of the state space X. Similarly, the N(k) multi-target measurement set at time k is represented as
(2)$$Z_{k}=\{z_{k}^{1},z_{k}^{2},\ldots ,z_{k}^{N(k)}\}\in F(Z)$$
where $z_{k}^{1},z_{k}^{2},\ldots ,z_{k}^{N(k)}$ represent the N(k) multi-target measurements and F(Z) represents all subsets of the state space Z. Traditional Bayesian recursion involves two steps: prediction and update. Firstly, the prior probability density $p(X_{k}|Z_{1:k-1})$ at time k is obtained according to the probability density $p(X_{k-1}|Z_{1:k-1})$ at time k−1. Then, the posterior probability density $p(X_{k}|Z_{1:k})$ is obtained using the measurement set $Z_{k}$ at time k. However, in the framework of random finite sets, the integration process is difficult to solve. Therefore, Ronald Mahler proposed a probability hypothesis density filter based on the random finite set to solve this problem, which uses probability hypothesis density to replace the previous probability density [28,29]. The probability hypothesis density is the first-order moment of the multi-target posterior probability density. Its definition in the random finite set is
(3)$$D(x)=E[\sum_{x\in X}^{}\delta_{X}(x)]=\int \sum_{x\in X}^{}\delta_{X}(x)p(X)\delta X$$
where $\delta_{X}(x)$ is the delta-Dirac mass located in x. Similar to the Bayesian recursion, the PHD filter involves a prediction step and an update step. Firstly, according to the probability hypothesis density $D_{k-1|k-1}(x|Z_{k})$ at time k−1, the PHD prediction at time k is
(4)$$D_{k|k-1}(x)=\gamma_{k|k-1}(k)+\int p_{S,k}(\xi )f_{k|k-1}(x|\xi )D_{k-1|k-1}(\xi )d\xi$$
where $p_{S,k}(\xi )$ is the probability that the target still survives at time k, $f_{k|k-1}(x|\xi )$ is the transition density of individual targets, and $\gamma_{k|k-1}(k)$ is the intensity function of the birth RFS at time k. Then, using the measurement set $Z_{k}$, the PHD update at time k is
(5)$$D_{k}(x)=(1-P_{D}(x))D_{k|k-1}(x)+\sum_{z\in Z_{k}}\frac{P_{D}(x)L_{z}(x)D_{k|k-1}(x)}{\lambda \cdot c(z)+\int P_{D}(x)L_{z}(x)D_{k|k-1}(x)dx}$$
where $P_{D}(x)$ is the detection probability; $L_{z}(x)$ is the likelihood of individual targets; λ⋅c(z) is the PHD of clutter at time k, where λ denotes the average number of clutter points per scan; and c(z) denotes the spatial distribution of each clutter point.

## 3. Multi-Feature Matching GM-PHD Filter

### 3.1. GM-PHD Filter

It is clear from (4) and (5) that the PHD recursion involves multiple integrals that have no closed-form solutions in general. Therefore, [16] proposed a method of using Gaussian mixture to realize the probability hypothesis density filter. This filter must simultaneously consider the following three assumptions.

*A.1:* Each target follows a linear Gaussian dynamical model and the sensor has a linear Gaussian measurement model [16], i.e.,
(6)$$f_{{}_{k|k-1}}(x|\xi )=N(x;F_{k-1}\xi ,Q_{k-1})$$
(7)$$g_{{}_{k|k-1}}(z|x)=N(z;H_{k}x,R_{k})$$
where N(•;m,P) denotes a Gaussian density with mean m and covariance P, $F_{k-1}$ is the state transition matrix, $Q_{k-1}$ is the process noise covariance, $H_{k}$ is the observation matrix, and $R_{k}$ is the observation noise covariance.

*A.2:* The survival and detection probabilities are state independent, i.e.,
(8)$$p_{S,k}(x)=p_{S,k}$$
(9)$$p_{D,k}(x)=p_{D,k}$$

*A.3:* The intensity of the birth RFS is Gaussian mixtures of the form. For simplicity, no spawning is considered here.

Compared to the PHD filter, the GM-PHD filter adds the pruning and merging procedures of Gaussian components in addition to the prediction and update. Assume that the probability hypothesis density at time k−1 is
(10)$$D_{k-1}(x)=\sum_{i=1}^{J_{k-1}}w_{k-1}^{i}N(x;m_{k-1}^{i},P_{k-1}^{i})$$
where $J_{k-1}$ represents the number of Gaussian components at time k−1 and $w_{k-1}^{i}$ represents the weight of the ith Gaussian component among all Gaussian components at time k−1. The PHD prediction at time k is
(11)$$D_{k|k-1}(x)=D_{S,k|k-1}(x)+\gamma_{k}(x)$$

In (11), $D_{S,k|k-1}(x)$ represents the predicted intensity of all survival targets at time k, i.e.,
(12)$$D_{S,k|k-1}(x)=p_{S,k}\sum_{i=1}^{J_{k-1}}w_{k-1}^{i}N(x;m_{S,k|k-1}^{i},P_{S,k|k-1}^{i})$$

In (12), $m_{S,k|k-1}^{i}$ and $P_{S,k|k-1}^{i}$ are respectively:(13)$$m_{S,k|k-1}^{i}=F_{k-1}m_{k-1}^{i}$$
(14)$$P_{S,k|k-1}^{i}=Q_{k-1}+F_{k-1}P_{k-1}^{i}F_{k-1}^{T}$$

In (11), $\gamma_{k}(x)$ represents the predicted intensity of all birth targets at time k, i.e.,
(15)$$\gamma_{k}(x)=\sum_{i=1}^{J_{k-1}}w_{\gamma ,k}^{i}N(x;m_{\gamma ,k}^{i},P_{\gamma ,k}^{i})$$

Therefore, the predicted intensity at time k is a Gaussian mixture of the form
(16)$$D_{k|k-1}(x)=\sum_{i=1}^{J_{k|k-1}}w_{k|k-1}^{i}N(x;m_{k|k-1}^{i},P_{k|k-1}^{i})$$
where $J_{k|k-1}$ represents the number of predicted Gaussian components at time k. Then, the measurement set $Z_{k}$ at time k is used to update the probability hypothesis density and the posterior probability hypothesis density can be obtained as follows:(17)$$D_{k}(x)=(1-P_{D,k})D_{k|k-1}(x)+\sum_{z\in Z_{k}}\sum_{i=1}^{J_{k|k-1}}w_{k}^{i}(z)N(x;m_{k|k}^{i}(z),P_{k|k}^{i}(z))$$
where $\left(1-P_{D,k})D_{k|k-1}(x\right)$ represents the PHD of misdetection targets. After an iterative process, the weight of the Gaussian component is
(18)$$w_{k}^{i}(z)=\frac{P_{D}(k)w_{k|k-1}^{i}q_{k}^{i}(z)}{\kappa_{k}(z)+P_{D}(k)\sum_{j=1}^{J_{k|k-1}}w_{k|k-1}^{i}q_{k}^{i}(z)}$$

In (18), the value of $q_{k}^{i}(z)$ is determined by how close the measurement set z is to the mean of the ith Gaussian component. The expression for $q_{k}^{i}(z)$ is
(19)$$q_{k}^{i}(z)=N(z;H_{k}m_{k|k-1}^{i},R_{k}^{i}+H_{k}P_{k|k-1}^{i}H_{k}^{T})$$
where
(20)$$m_{k|k}^{i}=m_{k|k-1}^{i}+K_{k}^{i}(z-H_{k}m_{k|k-1}^{i})$$
(21)$$K_{k}^{i}=P_{k|k-1}^{i}H_{k}^{T}{\left[H_{k}P_{k|k-1}^{i}H_{k}^{T}+R_{k}\right]}^{-1}$$
(22)$$P_{k|k}^{i}=(I-K_{k}^{i}H_{k})P_{k|k-1}^{i}$$

Next, a simple pruning procedure can be used to reduce the number of Gaussian components propagated to the next time step. A good approximation to the Gaussian mixture posterior intensity can be obtained by truncating components that have weak weights [16]. This can be done by discarding those with weights below the truncation threshold $T_{th}$. Moreover, some of the Gaussian components are so close together that they could be accurately approximated by a single Gaussian. Hence, in practice these components can be merged into one [16]. The purpose of pruning Gaussian components with weak weight is to prevent the number of Gaussian components from increasing indefinitely, and merging Gaussian components can reduce the number of Gaussian components to a certain extent so as to reduce the amount of computation. During the pruning and merging procedure, the truncation threshold $T_{th}$, merging threshold $U_{th}$, and maximum allowable number of Gaussian components $J_{\max}$ need to be selected with appropriate values. Suppose there are n Gaussian components that are merged into one. Then, the Gaussian components after pruning and merging are
(23)$${\tilde{w}}_{k}^{l}=\sum_{i=1}^{n}w_{k}^{\left(i\right)}$$
(24)$${\tilde{m}}_{k}^{l}=\frac{1}{{\tilde{w}}_{k}^{l}}\sum_{i=1}^{n}w_{k}^{\left(i\right)}m_{k}^{\left(i\right)}$$
(25)$${\tilde{P}}_{k}^{l}=\frac{1}{{\tilde{w}}_{k}^{l}}\sum_{i=1}^{n}w_{k}^{i}[P_{k}^{i}+({\tilde{m}}_{k}^{l}-m_{k}^{i}){\left({\tilde{m}}_{k}^{l}-m_{k}^{i}\right)}^{T}]$$

The final task is to extract multiple-target state estimates. A better alternative is to select the means of the Gaussians that have weights greater than some threshold of $w_{th}$, e.g., 0.5. The mean of the predicted and updated number of targets are
(26)$${\hat{N}}_{k|k-1}={\hat{N}}_{k-1}P_{S}(k)+\sum_{i=1}^{J_{\gamma ,k}}w_{\gamma ,k}^{i}$$
(27)$${\hat{N}}_{k}={\hat{N}}_{k|k-1}[1-P_{D}(k)]+\sum_{z\in Z_{k}}\sum_{i=1}^{J_{k|k-1}}w_{k}^{i}(z)$$

### 3.2. MFGM-PHD Filter

In the GM-PHD filtering algorithm, the weight of Gaussian component is calculated by only the position information. Therefore, this paper proposes a multi-feature matching GM-PHD filtering method for radar multi-target tracking. Radar echo contains range, amplitude, azimuth, Doppler frequency, and other information. In radar systems, Doppler frequency is the difference between the radar receiving frequency and transmitting frequency, which can be calculated by
(28)$$f_{d}=\frac{2f_{r}v_{r}}{c}$$
where $f_{d}$ is the Doppler frequency and $f_{r}$ is the frequency of radar. $v_{r}$ is the radial velocity of the target, which can be calculated by $v_{r}=v\cos\theta$ where θ is the angle between the velocity v and the radial direction. c is the speed of light. It is clear from (28) that the Doppler frequency is related to the radial velocity of the target. Although the variation range of radial velocity is not as small as the RCS of the target, the radial velocity of the target generally does not change drastically during the interval between two adjacent frames. As such, the Doppler frequency of the target does not change drastically and is also a relatively stable feature. The amplitude information is related to the RCS, position, and velocity of the target. In the interval between two adjacent frames, the RCS of targets generally does not change drastically. Moreover, the amplitude information can distinguish targets from clutter. Therefore, this paper uses Doppler frequency information and amplitude information to modify the weights of Gaussian components so as to improve tracking accuracy.

The traditional GM-PHD filter uses Equation (18) to update the weight of each Gaussian component. The variable affecting the weight of the Gaussian components is $q_{k}^{i}(z)$. Its expression is Equation (19). Equation (19) only contains position information; in order to update the weight of a Gaussian component with feature information, it is necessary to transform Equation (19). 

Equation (19) uses the Gaussian probability density function to describe the matching degree between the position of the measurement set and the position information contained in the predicted Gaussian component. Therefore, the Gaussian probability density function can also be used to describe the matching degree between the feature information contained in the measurement set and the feature information contained in the predicted Gaussian component. The expression of $q_{k}^{{}^{\prime }(i)}(z_{eq})$ after correction with feature information is
(29)$$q_{k}^{{}^{\prime }(i)}(z_{eq})=\frac{q_{k}^{i}(z)+q_{k}^{i}(f_{d})+q_{k}^{i}(a_{d})}{3}$$
where $z_{eq}$ represents the measurement set containing position information and feature information. $q_{k}^{i}(f_{d})$ and $q_{k}^{i}(a_{d})$ reflect the matching degree and predicted value of the Doppler frequency value and amplitude value contained in $z_{eq}$, respectively.
(30)$$q_{k}^{i}(f_{d})=N(f_{d};{\tilde{f}}_{d(k|k-1)},\sigma_{f_{d}}^{2})$$
(31)$$q_{k}^{i}(a_{d})=N(a_{d};{\tilde{a}}_{d(k|k-1)},\sigma_{a_{d}}^{2})$$
where $\sigma_{f_{d}}^{2}$ and $\sigma_{a_{d}}^{2}$ reflect the proportion of the matching degree of Doppler frequency value and amplitude value in calculating the total matching degree, respectively.

After replacing $q_{k}^{i}(z)$ in Equation (18) with $q_{k}^{{}^{\prime }(i)}(z_{eq})$, the new weight of Gaussian component is
(32)$$w_{k}^{{}^{\prime }(i)}(z)=\frac{P_{D}(k)w_{k|k-1}^{i}q_{k}^{{}^{\prime }(i)}(z_{eq})}{\kappa_{k}(z)+P_{D}(k)\sum_{j=1}^{J_{k|k-1}}w_{k|k-1}^{i}q_{k}^{{}^{\prime }(i)}(z_{eq})}$$

The new posterior intensity at time k is
(33)$$D_{k}(x)=(1-P_{D,k})D_{k|k-1}(x)+\sum_{z\in Z_{k}}\sum_{i=1}^{J_{k|k-1}}w_{k}^{{}^{\prime }(i)}(z)N(x;m_{k|k}^{i}(z),P_{k|k}^{i}(z))$$

Then, the Gaussian components are pruned and merged. When the Gaussian components are merged in addition to the position information of all the Gaussian components, the multi-feature information contained in all the Gaussian components are also merged. During the pruning and merging procedure, clutter can be further eliminated and different targets can be prevented from merging using Doppler and amplitude information.

Assuming that there are n Gaussian components to be merged, the total weight of the new Gaussian component after merging is
(34)$${\tilde{w}}_{k}^{{}^{\prime }(l)}=\sum_{i=1}^{n}w_{k}^{{}^{\prime }(i)}$$

The mean and covarience of the new Gaussian component after merging are
(35)$${\tilde{m}}_{k}^{{}^{\prime }(l)}=\frac{1}{{\tilde{w}}_{k}^{{}^{\prime }(l)}}\sum_{i=1}^{n}w_{k}^{{}^{\prime }(i)}m_{k}^{\left(i\right)}$$
(36)$${\tilde{P}}_{k}^{{}^{\prime }(l)}=\frac{1}{{\tilde{w}}_{k}^{{}^{\prime }(l)}}\sum_{i=1}^{n}w_{k}^{{}^{\prime }(i)}[P_{k}^{i}+({\tilde{m}}_{k}^{l}-m_{k}^{i}){\left({\tilde{m}}_{k}^{l}-m_{k}^{i}\right)}^{T}]$$

The Doppler frequency and amplitude after merging are respectively
(37)$${\tilde{f}}_{dk}^{{}^{\prime }(l)}=\frac{1}{{\tilde{w}}_{k}^{{}^{\prime }(l)}}\sum_{i=1}^{n}w_{k}^{{}^{\prime }(i)}f_{dk}^{\left(i\right)}$$
(38)$${\tilde{a}}_{dk}^{{}^{\prime }(l)}=\frac{1}{{\tilde{w}}_{k}^{{}^{\prime }(l)}}\sum_{i=1}^{n}w_{k}^{{}^{\prime }(i)}a_{dk}^{\left(i\right)}$$

After the pruning and merging procedure, multiple-target state estimates and feature information are extracted and up dated. Then, the above prediction step and update step are repeated.

## 4. Simulations and Results Analysis

### 4.1. Simulations

In this section, simulations are designed to demonstrate the efficiencies of the proposed MFGM-PHD filter. The results are compared with those of the standard GM-PHD filter [16], the joint-GLMB filter proposed by [19], and the LGM-PHD filter proposed by [20]. The simulations are performed as follows.

*Scenario**1:* For illustration purposes, consider a two-dimensional scenario with an unknown and time-varying number of targets observed in clutter over the surveillance region [−1000,1000] × [−1000,1000] (in m). The state $x_{k}={\left[x_{1,k},x_{2,k},x_{3,k},x_{4,k}\right]}^{T}$ of each target at time k consists of position ${\left[x_{1,k},x_{3,k}\right]}^{T}$ and velocity ${\left[x_{2,k},x_{4,k}\right]}^{T}$, while the measurement is a noisy version of the position. In the proposed MFGM-PHD filter, each target contains Doppler frequency and amplitude information ${\left[f_{dk},a_{dk}\right]}^{T}$ in addition to the state vector mentioned above. Each target moves according to the following linear Gaussian dynamics:(39)$$x_{k}=\left[\begin{array}{llll} 1 & T & 0 & 0\\ 0 & 1 & 0 & 0\\ 0 & 0 & 1 & T\\ 0 & 0 & 0 & 1 \end{array}\right]x_{k-1}+\sigma_{w}^{2}\left[\begin{array}{llll} T^{4}/4 & T^{3}/2 & 0 & 0\\ T^{3}/2 & T^{2} & 0 & 0\\ 0 & 0 & T^{4}/4 & T^{3}/2\\ 0 & 0 & T^{3}/2 & T^{2} \end{array}\right]$$
where T=1s is the sampling time and $\sigma_{w}=5{m/s}^{2}$ is the standard deviation of the process noise. Targets can appear and disappear in the scene at any time with the survival probability $p_{S,k}=0.99$ and detection probability $p_{D,k}=0.98$.

New targets appear spontaneously according to a Poisson point process with the intensity function
(40)$$\gamma_{k}(x_{k})=\sum_{i=1}^{3}w_{{}_{\gamma ,k}}^{\left(i\right)}N(x;m_{\gamma ,k}^{\left(i\right)},P_{\gamma ,k}^{\left(i\right)})$$
where $m_{\gamma ,k}^{\left(1\right)}={\left[100\;0\;400\;0\right]}^{T}$, $m_{\gamma ,k}^{\left(2\right)}={\left[450\;0\;300\;0\right]}^{T}$, $m_{\gamma ,k}^{\left(3\right)}={\left[100\;0\;150\;0\right]}^{T}$, $P_{\gamma ,k}^{\left(i\right)}=diag{\left({[225\;100\;225\;100]}^{T}\right)}^{2}$, $w_{\gamma ,k}^{\left(i\right)}=0.03$, and i=1,2,3. 

The target-originated measurements are given by
(41)$$y_{k}=\left[\begin{array}{l} 1\;0\;0\;0\\ 0\;0\;1\;0 \end{array}\right]x_{k}+\left[\begin{array}{l} w_{1,k}\\ w_{2,k} \end{array}\right]$$
where $\left\{w_{1,k}\right\}$ and $\left\{w_{2,k}\right\}$ are mutually independent zero-mean Gaussian white noise with standard deviations of $\sigma_{w1}=\sigma_{w2}=4$. Clutter is uniformly distributed in the surveillance region with an average rate of λ=60 points per scan. In the proposed MFGM-PHD filter, the Doppler frequency of three targets can be obtained by Equation (28) and the amplitude of each target is normal distribution with a mean of 0 and a variance of 10. $\sigma_{f_{d}}$ and $\sigma_{a_{d}}$ in Equations (30) and (31) are 0.85. In this paper, the parameters $T_{th}=10^{-5}$, $U_{th}=4$, and $J_{\max}=100$ are applied. One hundred Monte Carlo (100-MC) trials on the same target trajectory are performed to capture the average performance, but with independently generated observations for each trial. The results are presented in terms of the multi-target measure optimal sub-pattern assignment metric (OSPA) [30], the cardinality of the targets detected, and time costs.

Figure 2 shows the true target trajectories. Target 1 and target 2 are born at the same time, k = 1 s, but at two different locations. Target 3 is new target that appears at k = 10 s.

Figure 3 and Figure 4 show the trajectories of the targets as well as the tracking result of a single sample run of different filters. As can be seen from Figure 3 and Figure 4, when the detection probability is high and clutter density is low ($P_{D,k}=0.98,r=60$), the four filters can track well.

Figure 5 shows the mean OSPA distances versus time of different filters. Figure 6 shows the mean cardinality of the targets detected of different filters. It is clear that the mean OSPA distance and the mean cardinality of the proposed MFGM-PHD filter are better than that of other three filters.

Figure 7 shows the computing time of different filters. It shows that the computing time of the proposed MFGM-PHD filter is slightly better than that of the LGM-PHD filter, but obviously better than that of GM-PHD filter and joint-GLMB filter.

Therefore, the proposed MFGM-PHD filter is superior to GM-PHD filter and LGM-PHD filter in tracking performance and superior to joint-GLMB filter and GM-PHD filter in real-time performance.

*Scenario 2**:* In this example, we evaluate the performance of four filters with incrementing clutter rates. When the detection probability is $P_{D,k}=0.98$, the performances of the four filters are compared at clutter rates of λ=120, λ=180, λ=240, and λ=300.

Figure 8 shows the mean OSPA distances of different filters under various clutter rates. Figure 9 shows the mean cardinality of different filters under various clutter rates. The results show that as the clutter rate increases, the mean OSPA distances and the mean cardinality of the proposed MFGM-PHD filter are almost unchanged. The dense clutter environment affects the performance of other three filters.

Figure 10 shows the computing times of different filters under various clutter rates. It shows that as the clutter rate increases, the computing time of the joint-GLMB filter increases obviously, the average computing times of the proposed MFGM-PHD filter and LGM-PHD filter are basically unchanged. However, the tracking accuracy performance of the proposed MFGM-PHD filter is superior to the LGM-PHD filter. Compared to the other three filters, the proposed MFGM-PHD filter still has excellent real-time performance in a dense clutter.

In order to further visually display the average computing times of the four filters under different clutter rates, Table 1 shows the average computing time of different filters under various clutter rates. We can see that the proposed MFGM-PHD filter has the best real-time performance under different clutter rates and the LGM-PHD filter is the second best.

*Scenario 3:* In this example, we evaluate the performance of four filters with declining detection probability. When the clutter rate is λ=300, the performance of the four filters is compared at the detection probability of $P_{D,k}=1$, $P_{D,k}=0.95$, $P_{D,k}=0.9$ and $P_{D,k}=0.85$.

Figure 11 shows the mean OSPA distances of different filters under various probabilities of detection. Figure 12 shows the mean cardinality of different filters under various probabilities of detection. The results show that as detection probability declines, the performance of all four filters declines. 

Figure 13 shows the computing times of different filters under various probabilities of detection. It shows that as the detection probability declines, the computing time of the joint-GLMB filter increases, the average computing times of the proposed MFGM-PHD filter and LGM-PHD filter are basically unchanged, but the tracking accuracy performance of the proposed MFGM-PHD filter is superior to the LGM-PHD filter. Compared to the other three filters, the proposed MFGM-PHD filter still has excellent real-time performance with low detection probability.

In order to further visually display the average computing times of the four filters under different probabilities of detection, Table 2 shows the average computing time of different filters under various probabilities of detection. It can be seen that the proposed MFGM-PHD filter has the best real-time performance under different probabilities of detection and that the LGM-PHD filter is the second best. 

### 4.2. Results Analysis

#### 4.2.1. Filter Quality Analysis

In *Scenario 1*, when the detection probability is high and clutter density is low ($P_{D,k}=0.98,\lambda =60$), the performance of different filters is compared. As can be seen from Figure 3 and Figure 4, the four filters can track well. In Figure 5 and Figure 6, it can be seen that the mean OSPA distance of the proposed MFGM-PHD filter is similar to or slightly higher than that of the joint-GLMB filter. However, when the new target appears at k = 10 s and targets die at k = 70 s, the mean OSPA distance of the joint-GLMB filter increases suddenly. The mean cardinality of the proposed MFGM-PHD filter is better than that of the joint-GLMB filter, especially when the new target appears at k = 10 s. When new targets appear at k = 1 s and k = 10 s, the proposed MFGM-PHD filter can track stably, but the other three filters all miss targets temporarily.

In *Scenario 2*, when the detection probability is $P_{D,k}=0.98$, the performance of four filters is compared under various clutter rates. Figure 8 shows that as the clutter rate increases, the mean OSPA distances of the proposed MFGM-PHD filter and the joint-GLMB filter are almost unchanged, but the mean OSPA distances of the GM-PHD filter and the LGM-PHD filter obviously increase. With a clutter rate of λ=120, the mean OSPA distances of the proposed MFGM-PHD filter, the LGM-PHD filter, and the joint-GLMB filter are similar but they are all superior to that of GM-PHD filter. However, the mean OSPA distances of the proposed MFGM-PHD filter is superior to that of LGM-PHD filter and the joint-GLMB filter when the new target appears at k = 10 s and targets die at k = 70 s. When the clutter rate is λ=180, λ=240, and λ=300, the mean OSPA distance of the proposed MFGM-PHD filter is significantly better than that of the LGM-PHD filter and the GM-PHD filter. The mean OSPA distance of the proposed MFGM-PHD filter is similar to or slightly higher than that of the joint-GLMB filter. However, when the new target appears at k = 10 s and targets die at k = 70 s, the mean OSPA distance of the proposed MFGM-PHD filter is lower than that of the joint-GLMB filter. Figure 9 shows that the proposed MFGM-PHD filter is better than the other three filters in cardinality estimation when new targets appear at k = 1 s and k = 10 s. When the clutter rate increases, the cardinality estimation performance of the proposed MFGM-PHD filter basically remains unchanged, while the cardinality estimation performance of the other three filters all decreases, especially for the LGM-PHD filter.

In *Scenario 3*, when the clutter rate is λ=300, the performance of the four filters is compared under various detection probability. Figure 11 shows that as detection probability declines, the mean OSPA distances of four filters are increased. When the clutter rate is λ=300 and the detection probability is less than or equal to 0.95, the LGM-PHD filter and GM-PHD filter almost fail to track targets, especially the LGM-PHD filter. When the detection probabilities are 1 and 0.95, the mean OSPA distance of the proposed MFGM-PHD filter is similar to that of the joint-GLMB filter. However, when the new target appears at k = 10 s and targets die at k = 70 s, the mean OSPA distance of the joint-GLMB filter increases. When the detection probabilities are 0.9 and 0.85, the mean OSPA distance of the proposed MFGM-PHD filter is lower than that of the joint-GLMB filter. Figure 12 shows that as the detection probability declines, the cardinality estimation performance of the four filters declines, especially the LGM-PHD filter. If the detection probability is less than or equal to 0.95, the LGM-PHD filter cannot accurately estimate the number of targets. When new targets appear at k = 1 s and k = 10 s, the proposed MFGM-PHD filter can accurately estimate the number of targets while the other three filters all miss targets.

The simulation results of the three different scenarios demonstrate that the tracking performance of the proposed MFGM-PHD filter was better than that of other three filters in dense clutter and low detection probability environment because the proposed MFGM-PHD filter can distinguish targets from clutter using Doppler and amplitude information. Thus, the MFGM-PHD filter is more accurate to track multiple targets. The simulation results are coincident with the theory analysis.

#### 4.2.2. Computational Complexity Analysis

Figure 10 shows that as the clutter rate increases, the computing time of the joint-GLMB filter increases obviously, while the average computing times of the proposed MFGM-PHD filter and LGM-PHD filter are basically unchanged. The joint-GLMB filter sacrifices computing time to obtain greater tracking accuracy performance. However, in radar applications, real-time performance is as important as tracking accuracy. Compared to the other three filters, the proposed MFGM-PHD filter still has excellent real-time performance in dense clutter conditions.

Table 1 shows that the average computing time of the proposed MFGM-PHD filter is almost unaffected by the increase in clutter rate. However, when the clutter rate increases by a factor of five, the computing time of the joint-GLMB filter increases by a factor of seven and the GM-PHD filter increases by a factor of three. The advantage of the proposed MFGM-PHD filter can be fully demonstrated in a dense clutter environment.

Figure 13 shows that as the detection probability decreases, the computing time of the joint-GLMB filter increases, the average computing times of the proposed MFGM-PHD filter and LGM-PHD filter are basically unchanged, but the tracking accuracy performance of the proposed MFGM-PHD filter is superior to the LGM-PHD filter. 

Table 2 shows that the proposed MFGM-PHD filter still has excellent real-time performance with low detection probability.

Because the MFGM-PHD filter uses the Doppler information and amplitude information of the radar echo, the feature information can distinguish different targets. Thus, targets can be detected in a low detection probability environment and the advantage of the real-time performance can be fully demonstrated. The proposed MFGM-PHD filter can eliminate most clutter by use of the Doppler and amplitude information. As a result, the computing time is shortened. The simulation results are coincident with the theory analysis.

In summary, all simulation results show that the real-time performance of the MFGM-PHD filter is better than that of the GM-PHD filter, the LGM-PHD filter and the joint-GLMB filter. The proposed MFGM-PHD filter can adapt to high clutter density and low detection probability environments.

## 5. Conclusions

In this paper, we propose a multi-feature matching GM-PHD filtering method for radar multi-target tracking. Using the Doppler and amplitude information contained in the radar echo to modify the weights of Gaussian components, the target and clutter can be distinguished and the tracking accuracy and real-time performance can be improved. The results demonstrate that the performance of the proposed MFGM-PHD filter is better than the GM-PHD filter, LGM-PHD filter, and joint-GLMB filter not only in cardinality estimation and position estimation, but also in real-time performance with high clutter density and low detection probability conditions. However, some challenges still remain for the proposed filter, including the extended target tracking problem. Our focus in further studies is on the application of the proposed MFGM PHD filtering algorithm for extended target tracking.

## Figures and Tables

**Figure 1 sensors-22-05339-f001:**
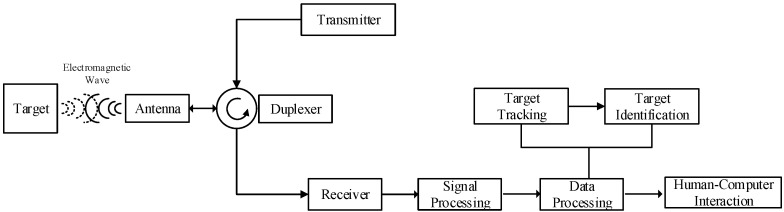
Simplified block diagram of radar system.

**Figure 2 sensors-22-05339-f002:**
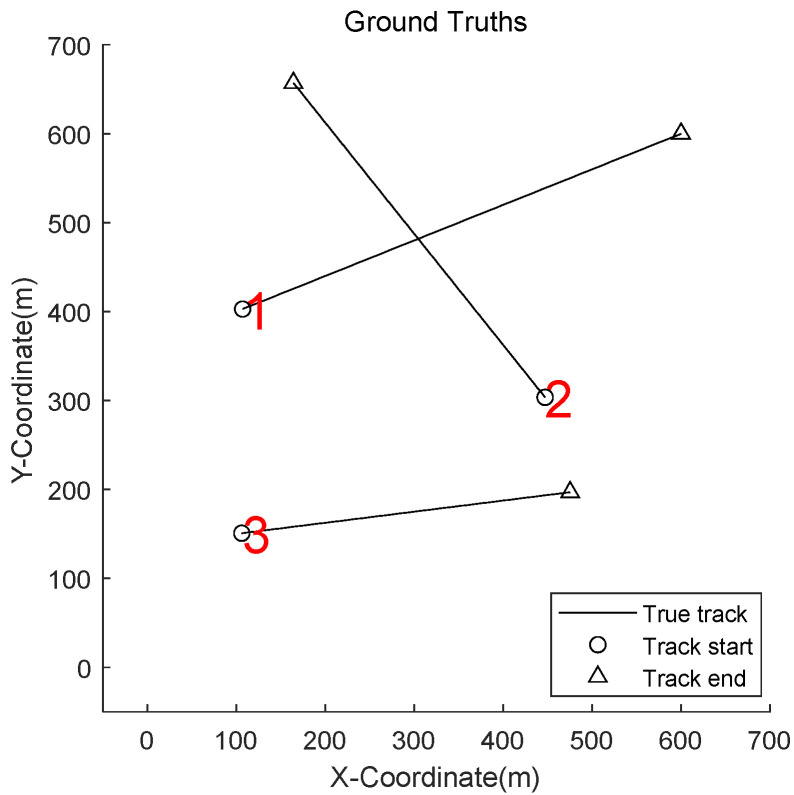
True trajectories of targets.

**Figure 3 sensors-22-05339-f003:**
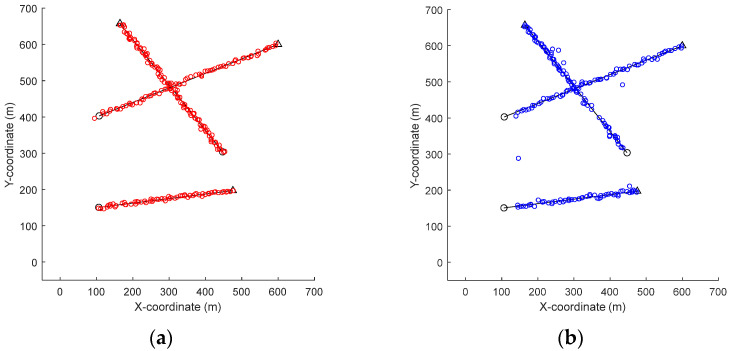
Ground truth trajectories of the targets (black) and estimated trajectories (colored) ($P_{D,k}=0.98,\lambda =60$). (**a**) MFGM–PHD filter (red). (**b**) GM–PHD filter (blue). (**c**) Joint–GLMB filter (yellow). (**d**) LGM–PHD filter (cyan).

**Figure 4 sensors-22-05339-f004:**
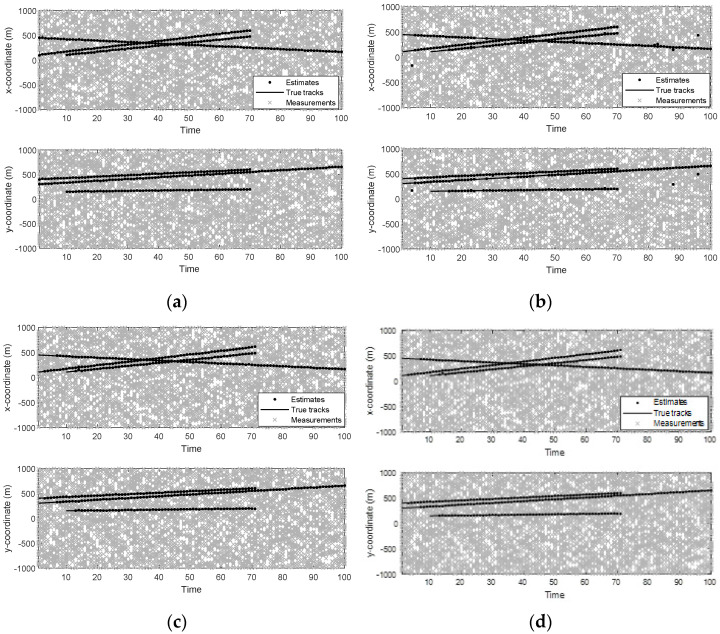
Position estimates of different filters ($P_{D,k}=0.98,\lambda =60$). (**a**) MFGM–PHD filter. (**b**) GM–PHD filter. (**c**) Joint–GLMB filter. (**d**) LGM–PHD filter.

**Figure 5 sensors-22-05339-f005:**
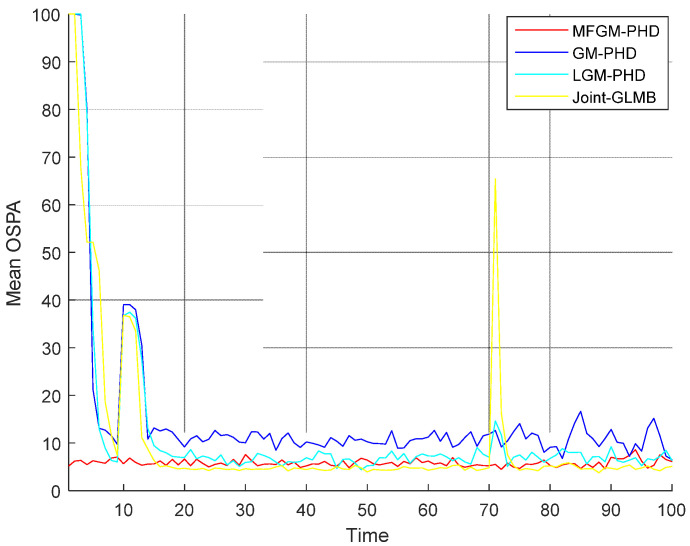
Mean OSPA distances versus time of different filters ($P_{D,k}=0.98,\lambda =60$).

**Figure 6 sensors-22-05339-f006:**
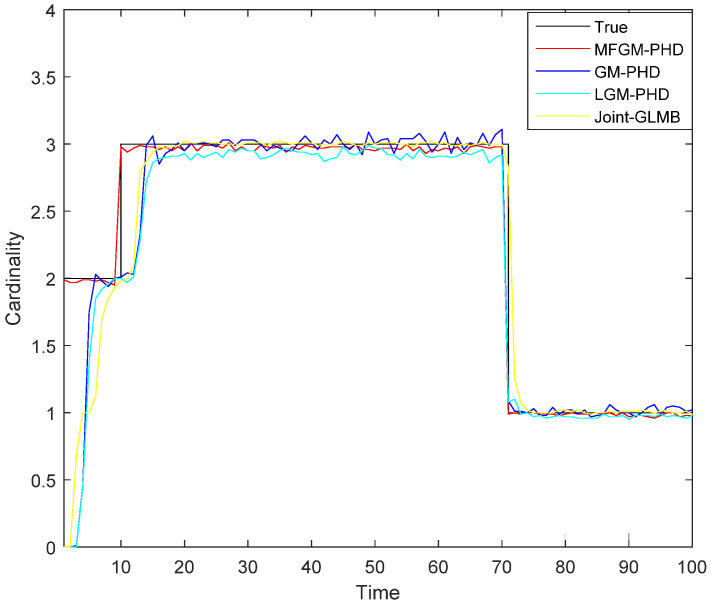
Mean cardinality of different filters ($P_{D,k}=0.98,\lambda =60$).

**Figure 7 sensors-22-05339-f007:**
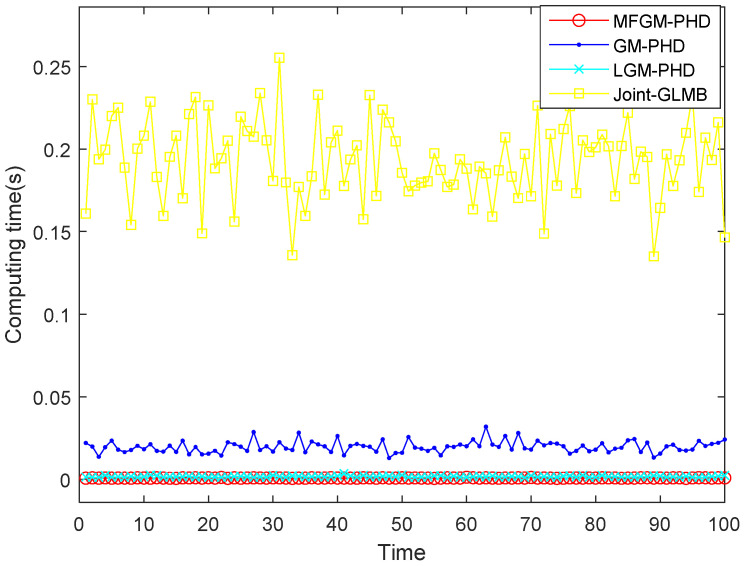
Computing time of different filters ($P_{D,k}=0.98,\lambda =60$).

**Figure 8 sensors-22-05339-f008:**
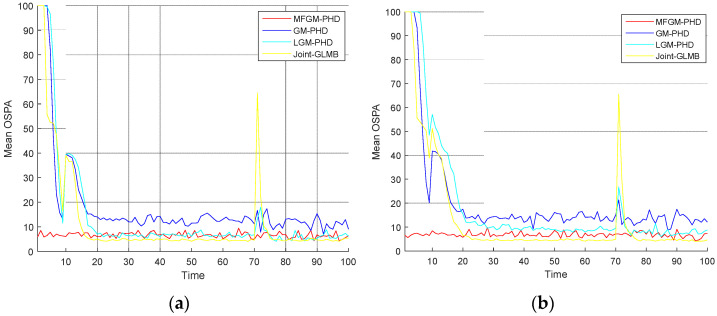
Mean OSPA distances of different filters under various clutter rates ($P_{D,k}=0.98$). (**a**) λ=120. (**b**) λ=180. (**c**) λ=240. (**d**) λ=300.

**Figure 9 sensors-22-05339-f009:**
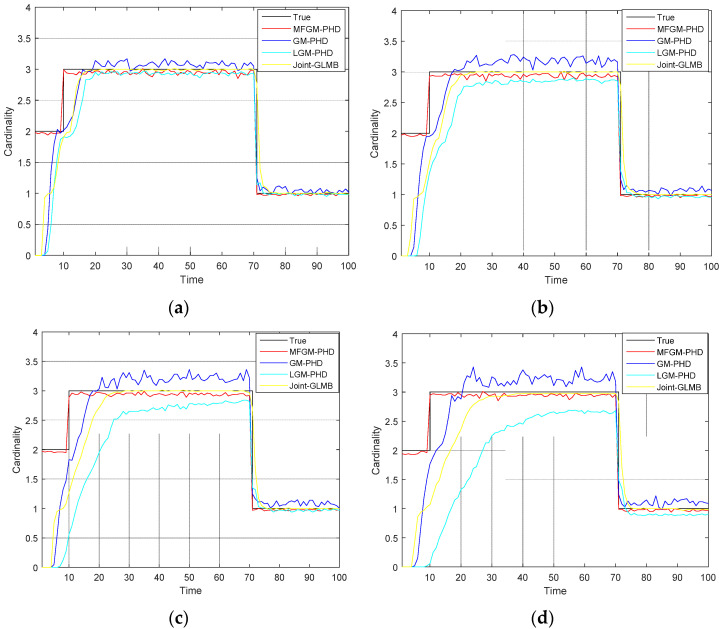
Mean cardinality of different filters under various clutter rates ($P_{D,k}=0.98$). (**a**) λ=120. (**b**) λ=180. (**c**) λ=240. (**d**) λ=300.

**Figure 10 sensors-22-05339-f010:**
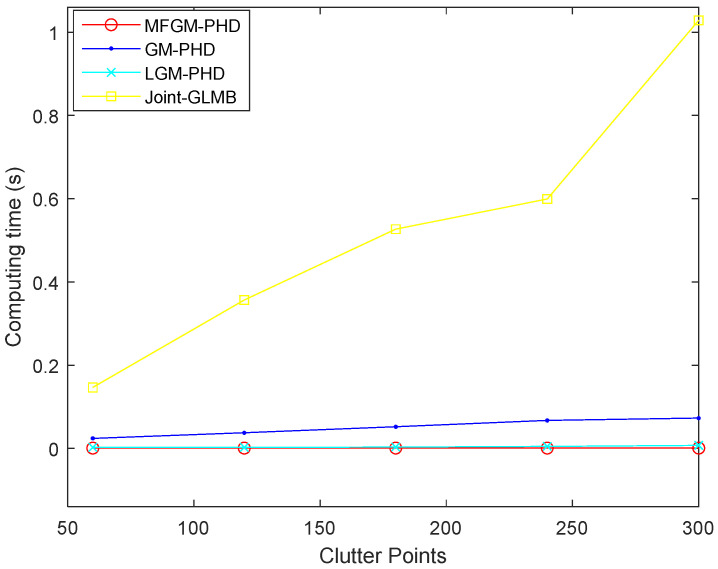
Computing times of different filters under various clutter rates ($P_{D,k}=0.98$).

**Figure 11 sensors-22-05339-f011:**
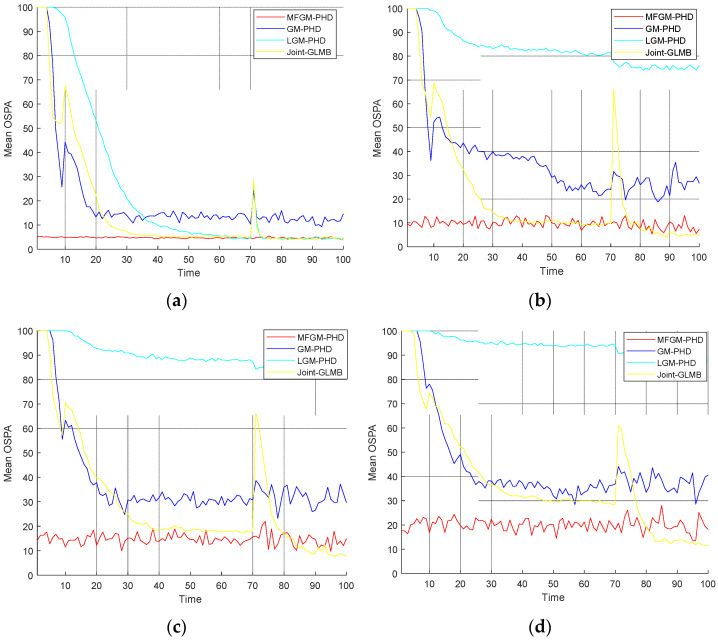
Mean OSPA distances of different filters under various probabilities of detection (λ=300). (**a**) $P_{D,k}=1$. (**b**) $P_{D,k}=0.95$. (**c**) $P_{D,k}=0.9$. (**d**) $P_{D,k}=0.85$.

**Figure 12 sensors-22-05339-f012:**
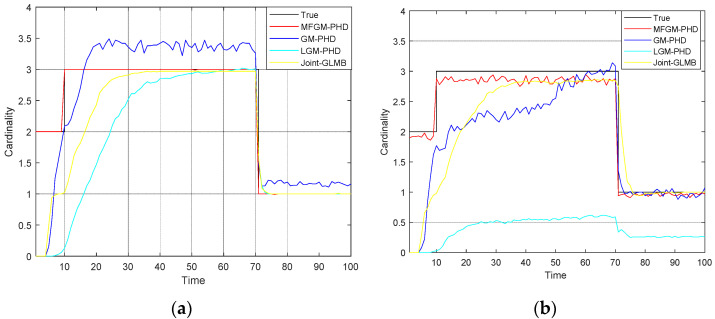
Mean cardinality of different filters under various probabilities of detection (λ=300). (**a**) $P_{D,k}=1$. (**b**) $P_{D,k}=0.95$. (**c**) $P_{D,k}=0.9$. (**d**) $P_{D,k}=0.85$.

**Figure 13 sensors-22-05339-f013:**
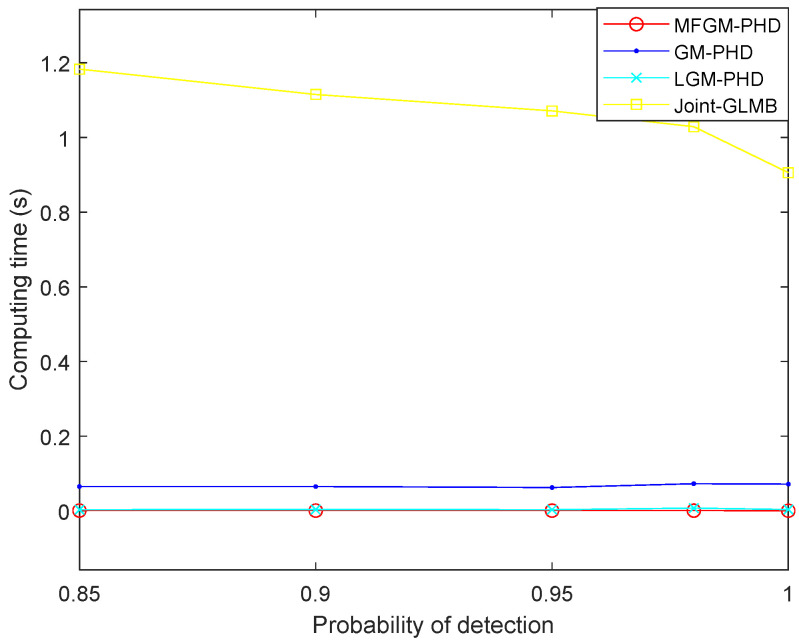
Computing times of different filters under various probabilities of detection (λ=300).

**Table 1 sensors-22-05339-t001:** The average computing time of different filters under various clutter rates ($P_{D,k}=0.98$ ).

Algorithms	Average Computing Time (s)				
	λ=60	λ=120	λ=180	λ=240	λ=300
MFGM-PHD	**0.000782**	**0.000803**	**0.000804**	**0.000812**	**0.000819**
GM-PHD	0.024075	0.037566	0.051871	0.067186	0.072739
LGM-PHD	0.002310	0.002337	0.003265	0.004861	0.007325
Joint-GLMB	0.146440	0.356536	0.526839	0.599240	1.028731

**Table 2 sensors-22-05339-t002:** The average computing time of different filters under various probabilities of detection (λ=300 ).

Algorithm	Average Computing Time (s)				
	$P_{D,k}=1$	$P_{D,k}=0.98$	$P_{D,k}=0.95$	$P_{D,k}=0.9$	$P_{D,k}=0.85$
MFGM-PHD	**0.000486**	**0.000819**	**0.000822**	**0.000827**	**0.000854**
GM-PHD	0.071858	0.072739	0.062451	0.065072	0.065333
LGM-PHD	0.003895	0.007325	0.003369	0.003493	0.00304
Joint-GLMB	0.905525	1.028731	1.071058	1.115119	1.182792

## Data Availability

Not applicable.

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
