# Peer review of "Multi-Feature Matching GM-PHD Filter for Radar Multi-Target Tracking"

_sensors, 2022, doi:10.3390/s22145339_

Round 1

Reviewer 1 Report

This paper tries to realize radar multi-target tracking based on multi-feature matching GM-PHD filter. This task is interesting and important. Extensive experiments show the effectiveness of the presented method. Although some of the modules adopted in the method are transferred from existing works, they do address some important issues. I am glad to recommend the publication of this paper if the following comments get carefully addressed.

1. The introduction of the article is not logical enough. The description of some references lacks clear explanation of motivation and contribution. And the description of motivation for this article is not enough logical and concise. The part of related work has the same problem.

2. What are the real difficulties that justify author’s work, i.e., what are the most important challenges authors want to handle? Why is it so difficult? I suggest to state this information clear in the introduction in order to give a better understand of the work.

3. To give a more intuitive comparison between the presented method and the state-of-the-art methods, more intermediate visual comparison results should appear in the main body of this paper.

4. Rather than simply stating the simulation results, I hope to see more relevant theoretical analysis in section 4.

5. Computational efficiency is critical to the implementation of the algorithm. Please add description about the algorithm complexity and compare with other methods.

6. I suggest to include reference citations for the comparison methods in the performance tables. Also, please underline the second best performance in the performance table for a better readability.

7. Feature representation is an important part of the proposed method. Some recent related work can be referred to, such as

https://doi.org/10.1016/j.isprsjprs.2021.12.004 and

https://doi.org/10.1109/TGRS.2022.3152250 .

8. The current version contains many grammar errors. The authors are suggested to carefully improve the presentation quality.

Author Response

Dear reviewer,

I am so appreciated your favorable comments on our manuscript entitled ''Multi-Feature Matching GM-PHD Filter for Radar Multi-Target Tracking''(sensors-1738737). Those comments are very helpful for revising and improving our paper, as well as the important guiding significance to other research. We have studied the comments carefully and made corrections which we hope meet with approval. According to your comments, we revised our paper carefully in the new manuscript. Please see the new manuscript.

Please see the attachment for detailed revisions according to each comment.

Please kindly email us if you have any questions.

Million thanks in advance!

                                                                                                    Mrs. Jin Tao

Reviewer 2 Report

The manuscripts presents the results of addition to algorithms of quite well developed area.  But the area of multi-target tracking is still very interesting for research because of a lot of new radar wave form signals development in recent years. After reading the manuscript some remarks arose that are needed to be clarified. Remarks 1. The random finite set (RFS) approach and The Gaussian Mixture Probability Hypothesis Density (GM-PHD) implementation were proposed more than 10 years ago, and the number of publications with various modifications of these methods, including in recent years, is large. The authors should have described in more detail the benefits and specifics of their proposed multi-feature matching GM-PHD (MFGM-PHD). 2. It is also useful to compare the characteristics of the proposed PHD filter not only with GM-PHD, but also with other filter modifications of this class described in the literature. 3. There is negligence in the supply of material: the axes of many drawings do not have designations, it is necessary to check the direction of the Duplexer arrow in Fig.1.

Author Response

Dear reviewer,

Thanks very much for taking your time to review our manuscript entitled ''Multi-Feature Matching GM-PHD Filter for Radar Multi-Target Tracking''(sensors-1738737). I really appreciate all your comments and suggestions. Those comments are very helpful for revising and improving our paper, as well as the important guiding significance to other research. We have studied the comments carefully and made corrections which we hope meet with approval. According to your comments, we revised our paper carefully in the new manuscript. Please see the new manuscript.

Please see the attachment for detailed revisions according to each comment.

Please kindly email us if you have any questions. Million thanks in advance!

Best regards

                                                                                                    Mrs. Jin Tao

Reviewer 3 Report

The comments are embedded in the attached PDF document.

Author Response

Dear reviewer,

Thanks very much for taking your time to review our manuscript entitled ''Multi-Feature Matching GM-PHD Filter for Radar Multi-Target Tracking''(sensors-1738737). I really appreciate all your comments and suggestions. Those comments are very helpful for revising and improving our paper. We have studied the comments carefully and made corrections in the new manuscript which we hope meet with approval. Because the new manuscript has changed a lot, we have also carefully checked and improved the English writing in the revised manuscript. Please see the new manuscript.

Please kindly email us if you have any questions. Million thanks in advance!

Thanks again for your favorable comments on our paper. Really appreciate!

Best regards

                                                                                                     Mrs. Jin Tao

Round 2

Reviewer 2 Report

Dear authors! Thanks a lot for your work under manuscript improvement. I'm enough satisfied with your answers. For future manuscripts I suggest you to provide more clear scientific novelty of proposed methods. 

Hope your fruitful future work!